# Constitutive Descriptions and Restoration Mechanisms of a Fe-17Cr Alloy during Deformation at Temperatures of 700–1000 °C

**DOI:** 10.3390/ma14175022

**Published:** 2021-09-03

**Authors:** Fei Gao, Zilong Gao, Qiyong Zhu, Zhenyu Liu

**Affiliations:** 1School of Materials Science and Engineering, Northeastern University, Shenyang 110819, China; 2Key Laboratory of Lightweight Structural Materials Liaoning Province, Northeastern University, Shenyang 110819, China; 3State Key Laboratory of Rolling and Automation, Northeastern University, Shenyang 110819, China; 1900494@stu.neu.edu.cn (Z.G.); 1970498@stu.neu.edu.cn (Q.Z.); zyliu@mail.neu.edu.cm (Z.L.)

**Keywords:** Fe-17Cr alloy, warm deformation, dynamic recrystallization mechanism, constitutive description

## Abstract

The deformation behavior for highly purified Fe-17Cr alloy was investigated at 700~1000 °C and 0.5~10 s^−1^. The microstructure evolution and corresponding mechanism during deformation were studied in-depth, using electron backscattering diffraction, transmission electron microscopy and precession electron diffraction. During deformation, dynamic recrystallization (DRX) occurred, along with extensive dynamic recovery, and the active DRX mechanism depended on deformation conditions. At higher Zener-Hollomon parameter (*Z* ≥ 5.93 × 10^27^ s^−1^), the development of the shear band was promoted, and then continuous DRX was induced by the formation and intersection shear band. At lower Zener-Hollomon parameter (*Z* ≤ 3.10 × 10^25^ s^−1^), the nucleation of the new grain was attributed to the combination of continuous DRX by uniform increase in misorientation between subgrains and discontinuous DRX by grain boundary bulging, and with increasing temperature, the effect of the former became weaker, whereas the effect of the latter became stronger. The DRX grain size increased with the temperature. For alleviating ridging, it seems advantageous to activate the continuous DRX induced by shear band through hot deformation with higher *Z*. In addition, the modified Johnson-Cook and Arrhenius-type models by conventional way were developed, and the modified Johnson-Cook model was developed, using the proposed way, by considering strain dependency of the material parameters. The Arrhenius-type model was also modified by using the proposed way, through distinguishing stress levels for acquiring partial parameter and through employing peak stress to determine the activation energy and considering strain dependency of only other parameters for compensating strain. According to our comparative analyses, the modified Arrhenius-type model by the proposed approach, which is suggested to model hot-deformation behavior for metals having only ferrite, could offer a more accurate prediction of flow behavior as compared to other developed models.

## 1. Introduction

Fe-Cr alloy steels are finding wide applications and replacing some Cr-Ni austenitic steels because of their excellent stress corrosion cracking resistance; high thermal conductivity; and low expansion coefficient and cost [1,2]. However, Fe-Cr alloys generally exhibit low formability and poor surface quality.

Improving these shortcomings for Fe-Cr alloys has been attempted by controlling rolling process [3,4,5,6,7,8,9]. Introducing the warm-rolling process is the effective way to increase the intensity or uniformity of γ-fiber recrystallization texture, obtain a more homogeneous distribution of oriented grain colonies in the final sheet, and eventually increase the *r*-value and weaken the occurrence of ridging by enhancing grain fragmentation and grain interior nucleation and refining hot-rolled and annealed microstructure. Dynamic recrystallization (DRX) holds great promise for controlling and refining the microstructure and thereby improving the *r*-value and alleviating ridging [10,11,12]. To further optimize the warm-rolling process, the systematic investigation of flow behavior and microstructure evolution, especially DRX behavior during warm deformation, is necessary.

Many efforts have been focused on understanding flow behavior and/or microstructure evolution of the Fe-Cr alloys with high stacking fault energy (SFE) [13,14,15,16,17,18,19,20]. It is well-known that, for some ferritic alloys with high SFE and austenitic alloys with low or medium SFE, continuous DRX (CDRX) and discontinuous DRX (DDRX) may occur at different deformation conditions, even at same condition [10,21,22]. Zhang et al. [14] and Liu et al. [15] found that, for highly purified Fe-21Cr or Fe-17Cr alloys, no new grains formed during deformation at 900 to 1150 °C. Kim and Yoo [19] proposed that, for Fe-17Cr alloy, CDRX was responsible for the deformation mechanism at 900 to 1100 °C, and the effect of CDRX enhanced with the increase of strain rate and the decrease of temperature, namely increasing the Zener-Hollomon parameter. Mehtonen et al. [18] suggested that, for stabilized Fe-21Cr alloy, extensive dynamic recovery (DRV) and CDRX occurred during hot deformation at 950 to 1050 °C, and CDRX was observed only under low Zener-Hollomon parameter conditions. Zu et al. [17] found that, for stabilized Fe-17Cr alloy, the CDRX had a prominent effect on microstructure evolution at low strain rate, whereas at a high strain rate, DDRX occurs during deformation at 850–1100 °C. It can be seen that the dynamic softening mechanisms and the corresponding kinetics for Fe-Cr alloys are still unclear and ambiguously, which are related to a lack of experimental data and different approaches to their interpretation. Moreover, these studies are concentrated on the deformation behavior at elevated temperatures. A very limited reference can only be provided for the basic research of microstructure–property relationships, such as DRX simulation and the optimization of the warm-rolling process (finish rolling at 800–700 °C).

In addition, regarding the flow behavior and constitutive models of Fe-Cr alloys, a majority of constitutive equations have been developed based on the flow behavior during hot deformation at elevated temperatures above 900 °C [16,17,18]. On the other hand, some crucial issues are generally ignored not only for Fe-Cr alloys, but also for most other materials. The constitutive equations for Fe-Cr alloys were usually constructed according to the phenomenological constitutive model. For the Arrhenius-type models, owing to the fact that the effect of strain on flow behavior has not been considered, it was generally modified by incorporating strain-dependent material parameters for compensating the effect of strain in Fe-Cr alloys [16,17,18], which were applied for other alloys [23,24] and steels [25,26,27]. Nevertheless, there are two following issues with the abovementioned method: (1) One issue is that the effect of strain is compensated by considering all parameters described in Equation (6) in Section 3.2.3. as the polynomial function of strain. It is well-known that, at a given strain and different temperatures and strain rates, flow curves may be in different stages of deformation and samples may take place different deformation or softening mechanisms, leading to the formation of different microstructure characteristic and corresponding flow stresses. Therefore, the abovementioned consideration about the strain will result in the phenomenon that, through utilizing the flow stress obtained according to the different microstructure characteristic (e.g., different stages of dynamic softening or different substructures), the parameters (e.g., activation energy) at a certain strain are obtained. (2) Another issue is that the *α* (*≈β/n*′) is usually acquired through utilizing all experimental results for obtaining *n*′ in Equation (7), which is invalid for the experimental results with higher stress, and *β* in Equation (8), which is not suitable for the experimental results with lower stress, as described in in Equations (6)–(8) in Section 3.2.3. For the Johnson-Cook (JC) models, an important problem that needs to be addressed is that, generally, the material parameters *B*_5_, *λ*_1_ and *λ*_2_ (as stated in Equation (2) in this study) are all considered to be constants. As a matter of fact, according to studies about the establishment of JC model [16,28,29,30,31,32], these material parameters obtained by linear fitting are markedly different at different strains and are related to strains. This is to say, the conventional approach could introduce some errors into the parameters, which will decrease the reliability and stability of the developed constitutive model.

In this study, the deformation behavior of a highly purified Fe-17Cr alloy was investigated at temperature with a wide range which covers the practical deformation conditions during the warm-rolling process. The corresponding dynamic microstructure was analyzed in-depth, using electron backscatter diffraction (EBSD), transmission electron microscopy (TEM) and precession electron diffraction (PED). The constitutive description on the basis of the above models was constructed through utilizing conventional way and new way for solving the existing issues, and their performance was comparatively analyzed in detail and consequently a most suitable way was worked out. Moreover, the microstructure evolution was studied, and the corresponding mechanism was explored during deformation, with an objective to provide a theoretical basis for the basic research of microstructure–property relationships and the optimization of rolling process, especially the warm-rolling process.

## 2. Materials and Methods

The cast ingot of a highly purified Fe-17Cr alloy (0.004 C, 0.005 N, 0.07 Mn, 0.14 Si, 0.18 Nb, 17.2 Cr and the balance Fe, all in mass%) was hot-rolled to band with the thickness of 12 mm in the temperature range of 1100 to 900 °C after treating at 1150 °C for 1 h. Using a MMS 300 thermo-mechanical simulator (State Key Laboratory of Rolling and Automation, Northeastern University, Shenyang, China), the isothermal hot compression test was carried out by using the cylindrical sample (Φ8 mm × 15 mm) according to the deformation processes (Figure 1). Note that the axis of the sample is parallel to the rolling direction of the rolled band. During hot-compression tests, the loading direction is parallel to rolling direction, and two specimens were tested for each condition. The samples were firstly heated to 1150 °C at the 20 °C·s^−1^ heating rate and then held at this temperature for 180 s to homogenize temperature. They were then cooled to 700, 750, 800, 850, 900, 950 and 1000 °C at the 10 °C·s^−1^ cooling rate and held at corresponding temperatures for 30 s to form the uniform distribution of temperature. Then these samples were deformed at the 0.5, 1, 5 and 10 s^−1^ strain rates with the 0.55 strain. Tantalum foils and high-temperature lubricant (MoS_2_) were employed to minimize the friction coefficient during isothermal hot-compression tests, ensuring the reliability of measured data. The microstructure analyses were carried out by applying EBSD, TEM and PED, and the samples were prepared by applying the methods reported in Reference [33].

## 3. Results and Discussion

### 3.1. Flow Behavior

All the flow curves under different processes possess similar characteristics (Figure 2) that are in agreement with those of other steels with extensive DRV and DRX during deformation [18,34]. The flow-stress curves are typically composed of two regions, i.e., work hardening region and steady-state region. The flow stress decreases with increasing deformation temperature and decreasing strain rate. This can be attributed to the following two factors: (1) strengthening thermal vibration of atoms and decreasing binding force among atoms due to the increase of temperature, leading to low critical shear stress, high lattice defect mobility and corresponding low dislocation density; and (2) the weaker dislocation–dislocation interaction and longer dynamic softening time associated with the lower strain rate, resulting in the low dislocation density [20]. Moreover, the effect of strain rate on flow stress becomes weaker with a decreasing deformation temperature. The work hardening tendency with the change of strain rate can be exhibited by the strain rate sensitivity *m*, which can be evaluated by the slope in $\ln\sigma -\ln\dot{\varepsilon }$ plot through liner fitting [35]. With decreasing temperature, the slope in $\ln\sigma -\ln\dot{\varepsilon }$ plot decreases (Figure 3), indicating that the *m* decreases and the effect of strain rate on flow stress becomes weaker. This can be explained in term of the stacking fault energy. A similar sensitivity of the effect of strain rate on flow stress to deformation temperature has been observed and proved in other stainless steels [28,36].

### 3.2. Comparative Analysis of Constitutive Models

It is well-known that high strain rates may cause adiabatic heating in the specimens. The adiabatic temperature increase can be estimated from the flow curves employing the Taylor–Quinney coefficient which reflects the ratio of plastic work transformed into heat or stored energy. However, this coefficient is inconstant during deformation and its value is dependent of not only the amount of strain, but also the loading mode, strain rate and temperature at which the strain was achieved [37]. It was found that, during high rate deformation of transformation-induced plasticity steel, the Taylor–Quinney coefficient of 0.82~1.00 is observed [38]. Thus, the adiabatic temperature rise is extremely complication during deformation, especially at high rate, and its accurate calculation is extremely difficult. Extensive work is required in the systematic determination of the amount of plastic work that is transformed into heat in the metals [38]. On the other hand, the isothermal hot-compression tests are employed to investigate the deformation behavior in this study, and during the compression, the temperature is controlled by a thermo-mechanical simulator. When the temperature of specimen increases, due to adiabatic heating, there is still some time for adjusting the temperature to minimizing its temperature change as much as possible for the thermo-mechanical simulator, especially during deformation at the strain rate below 10 s^−1^. Overall, the actual temperature change is an extremely complicated subject during deformation, especially at high rate, and its effect on flow stress becomes overwhelmingly difficult to estimate correctly. Despite that, the numerical evaluation of the actual temperature rise is often simplified by only using the Taylor–Quinney coefficient as a constant value, which can lead to distinct overestimation of the adiabatic heating [37]. Therefore, the effect of adiabatic heating at high strain rate (10 s^−^^1^) is not considered during constitutive modeling.

#### 3.2.1. Modified Johnson-Cook Model

The JC model describing the flow behavior of metals can be expressed as follows [39]:(1)$$\sigma =\left(A_{1}+A_{2}\varepsilon^{n1}\right)\left[1+A_{3}\ln{\left(\frac{\dot{\varepsilon }}{{\dot{\varepsilon }}_{r}}\right)}^{}\right]\left[1-{\left(\frac{T-T_{r}}{T_{m}-T_{r}}\right)}^{n2}\right]$$
where the means of *σ*, *ε*, $\dot{\varepsilon }$, ${\dot{\varepsilon }}_{r}$, *T*, *T_r_*, *T_m_*, *A*_2_, *n*_1_, *A*_3_, *n*_2_ and *A*_1_ are identical to those reported in Reference [28]. Clearly, a coupled effect of temperature and strain rate on the flow stress, which is observed in Section 3.1, is not considered in this model. Liu et al. [29] revised the JC model for addressing this problem, which is expressed as follows:(2)$$\sigma =\left(B_{1}+B_{2}\varepsilon +B_{3}\varepsilon^{2}+B_{4}\varepsilon^{3}\right)[1+B_{5}\ln(\frac{\dot{\varepsilon }}{{\dot{\varepsilon }}_{r}})]\exp\left\{\left[\lambda_{1}+\lambda_{2}\ln\left(\frac{\dot{\varepsilon }}{{\dot{\varepsilon }}_{r}}\right)\right]\left(T-T_{r}\right)\right\}$$
where *λ*_1_, *λ*_2_, *B*_1_, *B*_2_, *B*_3_, *B*_4_ and *B*_5_ are parameters. This model has been utilized to predict flow behavior of some steels [29,30] and alloys [40,41]. Note that this modified Johnson-Cook model also considers the yield and strain hardening portion of the original Johnson-Cook model [29], so the equation for the flow behavior at reference conditions is also changed. In this section, the modified JC model is developed by conventional approach employed by other researchers [29,30].

The parameters *B*_1_, *B*_2_, *B*_3_, *B*_4_, *B*_5_, *λ*_1_ and *λ*_2_ at reference conditions of 700 °C and 0.5 s^−1^ are acquired to be 144 MPa, 920 MPa, −2497 MPa, 2266 MPa, 0.011, −0.00348 and 0.00029 by linear fitting, respectively, as shown in Appendix A. Subsequently, the parameters for other reference temperatures and strain rates are determined. By using these parameters, the predicted flow stresses can be acquired and eventually be employed for determining the different average absolute relative errors (*AARE*s) under different reference conditions (Appendix A) which can be expressed as follows:(3)$$AARE\left(\%\right)=\frac{1}{N}\sum_{i=1}^{N}\left|\frac{M_{i}-P_{i}}{M_{i}}\right|\times 100$$
where *M*, *P* and *N* are identical to those reported in Reference [28]. At 850 °C and 1 s^−1^, the minimum *AARE* of 3.4% is obtained.

Finally, the optimum parameters are obtained and shown in Table 1, and the modified JC model by conventional approach is shown by using the equation below:(4)$$\sigma =\left(103+439\varepsilon -1171\varepsilon^{2}+1019\varepsilon^{3}\right)\left[1+0.071\ln\left(\frac{\dot{\varepsilon }}{1}\right)\right]\exp\{[-0.00342+2.3\times 10^{-4}\ln\left(\frac{\dot{\varepsilon }}{1}\right)]\left(T-1123.15\right)\}$$

#### 3.2.2. Modified Johnson-Cook Model by Considering Strain Dependency of the Parameters

Nevertheless, in the abovementioned conventional approach, an important problem is that the material parameters *B*_5_, *λ*_1_ and *λ*_2_ are considered as constants. In fact, the *B*_5_, *λ*_1_ and *λ*_2_ are related to strains (Appendix A), and this phenomenon has also been observed in the establishing of JC model for other materials [16,31,32], as stated in Section 1. For overcoming this problem, the *B*_5_, *λ*_1_ and *λ*_2_ are considered as strain-dependent parameters by regarding these parameters as polynomial function of strain in this section. Note that the reference condition of 850 °C and 1 s^−1^ is employed because it is the most suitable reference condition according to Section 3.2.1.

(1)Determination of material parameters *B*_1_, *B*_2_, *B*_3_ and *B*_4_

Obviously, the *B*_1_, *B*_2_, *B*_3_ and *B*_4_ are 103, 439, −1171 and 1019 MPa at the reference condition of 850 °C and 1 s^−1^ according to Section 3.2.1.

(2)Determination of strain-dependent material parameters *B*_5_, *λ*_1_ and *λ*_2_

The *B*_5_, *λ*_1_ and *λ*_2_ at the strain of 0.3 are acquired to be 0.060, −0.00349 and 0.00021 by linear fitting, respectively (Appendix A). Subsequently, the *B*_5_, *λ*_1_ and *λ*_2_ for other strains are determined, and the strain-dependent parameters *B*_5_, *λ*_1_ and *λ*_2_ are perfectly suited to be expressed by 6th-order polynomial fitting (Appendix A). Finally, the modified JC model with partial strain-dependent parameters is shown by using the equation below:(5)$$\left\{\begin{array}{l} \sigma =\left(103+439\varepsilon -1171\varepsilon^{2}+1019\varepsilon^{3}\right)\left[1+B_{5}\ln\left(\frac{\dot{\varepsilon }}{1}\right)\right]\exp\left\{\left[\lambda_{1}+\lambda_{2}\ln\left(\frac{\dot{\varepsilon }}{1}\right)\right]\left(T-1123.15\right)\right\}\\ B_{5}=-0.13+5\varepsilon -46\varepsilon^{2}+215\varepsilon^{3}-537\varepsilon^{4}+689\varepsilon^{5}-357\varepsilon^{6}\\ \lambda_{1}=-0.00186-0.035\varepsilon +0.31\varepsilon^{2}-1.4\varepsilon^{3}+3.5\varepsilon^{4}-5\varepsilon^{5}+2.4\varepsilon^{6}\\ \lambda_{2}=6.0867\times 10^{-5}+0.008\varepsilon -0.10\varepsilon^{2}+0.6\varepsilon^{3}-1.7\varepsilon^{4}+2\varepsilon^{5}-1.4\varepsilon^{6} \end{array}\right.$$

#### 3.2.3. Modified Arrhenius-Type Model by Conventional Approach

The Arrhenius-type model describing the flow behavior of metals can be expressed as follows [42]:(6)$$\dot{\varepsilon }=A{\left[\sin h\left(\alpha \sigma \right)\right]}^{n}\cdot \exp\left(-\frac{Q}{RT}\right)\;(\mathrm{for}\;\mathrm{all}\;\mathrm{stress}\;\mathrm{levels})$$
(7)$$\dot{\varepsilon }=A^{\prime }\sigma^{n^{\prime }}\cdot \exp\left(-\frac{Q}{RT}\right)\;(\mathrm{for}\;\mathrm{low}\;\mathrm{stress}\;\mathrm{level}\;(\alpha \sigma \;<\;0.8))$$
(8)$$\dot{\varepsilon }=A^{\prime \prime }\exp\left(\beta \sigma \right)\cdot \exp\left(-\frac{Q}{RT}\right)\;(\mathrm{high}\;\mathrm{stress}\;\mathrm{level}\;(\alpha \sigma \;>\;1.2))$$
where the means of *A*, *n*, *A*′, *n*′, *A*″ and *β*, *α*, *Q* and *R* are identical to those reported in Reference [28]. In addition, the Zener-Holloman parameter *Z* describing the effects of temperature and strain rate on flow stress is shown by using the equation below [40]:(9)$$Z=\dot{\varepsilon }\cdot \exp\left(-\frac{Q}{RT}\right)$$

In this model, the effect of strain on flow stress has not been considered. As stated in Section 1, it is recently modified by considering the *Q*, *α*, *A* and *n* as polynomial function of strain. Moreover, *n*′ and *β* are determined the through use of all measured stresses, and then *α* (*≈β/n*′) is determined. In this section, this conventional approach is employed for developing modified Arrhenius-type model.

The *n*′, *β*, *α*, *Q*, *n* and ln*A* at the strain of 0.3 are acquired to be 22.47, 0.12 MPa^−1^, 0.0053 MPa^−1^, 471,332.541 J·mol^−1^, 11.0 and 51.4 by linear fitting, respectively, as shown in Appendix A. Subsequently, the *n*′, *β*, *α*, *Q*, ln*A* and *n* for other strains are calculated, and the strain-dependent parameters *α*, *Q*, *n* and ln*A* are perfectly suited to be expressed by 6th-order polynomial fitting, as shown in Appendix A. The Equations (6) and (9) can be expressed as follows [14]:(10)$$\sigma =\frac{1}{\alpha }\ln\left\{{\left(\frac{Z}{A}\right)}^{1/n}+{\left[{\left(\frac{Z}{A}\right)}^{2/n}+1\right]}^{1/2}\right\}$$

Finally, the modified Arrhenius-type model by conventional approach for the compensation of strain and the determination of key material parameter is shown by using the equation below:(11)$$\left\{\begin{array}{l} \sigma =\frac{1}{\alpha }\ln\left\{{\left(\frac{Z}{A}\right)}^{1/n}+{\left[{\left(\frac{Z}{A}\right)}^{2/n}+1\right]}^{1/2}\right\}\\ Z=\dot{\varepsilon }\cdot \exp\left(\frac{Q}{RT}\right)\\ \alpha =0.014-0.20\varepsilon +2.0\varepsilon^{2}-10\varepsilon^{3}+27\varepsilon^{4}-37\varepsilon^{5}+20\varepsilon^{6}\\ Q=1256-21,653\varepsilon +210,505\varepsilon^{2}-1,002,920\varepsilon^{3}+2,522,730\varepsilon^{4}-3,194,300\varepsilon^{5}+1,600,200\varepsilon^{6}\\ n=35-619\varepsilon +5881\varepsilon^{2}-27,794\varepsilon^{3}+69,947\varepsilon^{4}-89,266\varepsilon^{5}+45,370\varepsilon^{6}\\ \ln A=125-2015\varepsilon +19,290\varepsilon^{2}-90,440\varepsilon^{3}+223,932\varepsilon^{4}-278,814\varepsilon^{5}+137,075\varepsilon^{6} \end{array}\right.$$

#### 3.2.4. Modified Arrhenius-Type Model by Proposed Approach

However, two important problems are usually neglected by the abovementioned conventional approach. One problem is that, through utilizing the flow stress obtained according to the different microstructure characteristic (e.g., different stages of dynamic softening or different substructures), the parameters (e.g., activation energy) at a certain strain are obtained, because, at a given strain and different temperatures and strain rates, flow curves may be in different stages of deformation, and samples may take different deformations or softening mechanisms, leading to the formation of different microstructure characteristic and corresponding flow stresses. It is well-known that nearly the same deformation or softening stages can be represented by the characteristic stresses, such as peak stress at different deformation conditions. To solve the above issue, the determination of *Q* and *α* is suggested through utilizing the characteristic stress, peak stress *σ*_p_, which does not suffer from the differences in the microstructure characteristic at different deformation conditions, and the determination of *A* and *n* are suggested through considering them as polynomial functions of strain. This section employs this method, compensating the strain.

Another problem is that the *α* (*≈β/n*′) is usually acquired through utilizing all experimental results for obtaining *n*′ in Equation (7), which is invalid for the experimental results with higher stress, and *β* in Equation (8), which is not suitable for the experimental results with lower stress. To solve the above issue, distinguishing experimental results is suggested according to the flow-stress levels, and the determination of *n*′ and *β* is suggested through employing experimental results with low flow stresses and high flow stresses, respectively. This section employs this method, acquiring crucial parameter *α.*

Firstly, by utilizing peak stresses, we calculate the parameters *α* and *Q*. Based on deformation behavior of other Fe-Cr alloys reported by researchers [14,16,17], the *α* of the Fe-17Cr alloy is initially estimated as 0.009 MPa^−1^ for judging high peak stress and low peak stress and subsequently calculating *n*′ and *β*. In this case, the *n*′ is obtained by using the peak stresses during deformation at the conditions of 1000 °C and 0.5–10 s^−1^, and *β* is calculated by using the peak stresses during deformation at the conditions of 700–850 °C and 0.5–10 s^−1^. The *n*′, *β*, *α* and *Q* are acquired to be 12.68, 0.14 MPa^−1^, 0.0110 MPa^−1^ and 550,059.9 J·mol^−1^ by linear fitting, as shown in Appendix A.

Secondly, the parameters *A* and *n* are considered as the strain-dependent parameters. The ln*A* and *n* at the strain of 0.3 are acquired to be 50.3 and 8.3 by linear fitting, respectively, and those for other strains are also obtained, and the strain-dependent parameters *n* and ln*A* are perfectly suited to be expressed by 6th-order polynomial fitting, as shown in Appendix A. Finally, the modified Arrhenius-type model by the proposed approach is shown by using the equation below:(12)$$\left\{\begin{array}{l} \sigma =\frac{1}{0.0110}\ln\left\{{\left(\frac{Z}{A}\right)}^{1/n}+{\left[{\left(\frac{Z}{A}\right)}^{2/n}+1\right]}^{1/2}\right\}\\ Z=\dot{\varepsilon }\cdot \exp\left(\frac{550,059.9}{RT}\right)\\ n=19.5-222\varepsilon +1938\varepsilon^{2}-8988\varepsilon^{3}+22,846\varepsilon^{4}-30,006\varepsilon^{5}+15,908\varepsilon^{6}\\ \ln A=54.4-68\varepsilon +587\varepsilon^{2}-2874\varepsilon^{3}+7788\varepsilon^{4}-10,750\varepsilon^{5}+5905\varepsilon^{6} \end{array}\right.$$

#### 3.2.5. Evaluation of Constitutive Descriptions

For evaluating the constitutive descriptions, the predicted flow stresses are compared with the experimental data, as shown in Appendix A. It can be seen that, despite some errors observed at higher strain rates and lower temperatures (700 °C and 5 s^−1^; 700 °C and 10 s^−1^), the predicted values based on the constitutive descriptions in this study, especially the modified Arrhenius-type model by the proposed approach, can track the experimental data very well. The factors that generate the abovementioned deviations are as follows:(1)The deviations in the measurement of flow stress;(2)The deviations from data fitting for determining the parameters such as liner fitting;(3)The limitation of the phenomenological constitutive model [43,44,45,46];(4)The adiabatic temperature rises during high rate-deformation heating [47]. It should be noted that only a very small amount of the flow stress is affected by temperature rise and corresponding thermal softening (such as at 700 °C and 10 s^−1^). This has a weak effect on the relationships between flow stress and deformation conditions and the determination of material parameters; that is to say, only a small part of thermal softening effect can be considered during the constitutive modeling. Therefore, the flow stresses predicted by the developed constitutive models are still overestimated at some deformation conditions; however, the experimental data affected by thermal softening were also used for the constitutive modeling. The combined effect of abovementioned factors (1)–(4) results in the deviations, so these deviations are not evenly distributed around the experimental curve.

In order to quantitatively evaluate the developed constitutive models, the correlation coefficient (*R*) and average absolute relative error (*AARE*) are employed. The *R* is shown in the equation below:(13)$$R=\frac{\sum_{i=1}^{N}\left(M_{i}-\bar{M})(P_{i}-\bar{P}\right)}{\sqrt{\sum_{i=1}^{N}{\left(M_{i}-\bar{M}\right)}^{2}\sum_{i=1}^{N}{\left(P_{i}-\bar{P}\right)}^{2}}}$$
where the means of *M*, *P*, $\bar{M}$, $\bar{P}$ and *N* are identical to those reported in Reference [33]. The measured and predicted data for the modified Arrhenius-type model by the proposed approach exhibit a better correlation as compared to the other three developed constitutive models (Figure 4). Moreover, the *AARE* for the modified Arrhenius-type models by the proposed approach is also lower as compared to the other three developed constitutive models, as shown in Figure 4. For further evaluating the developed constitutive models, the isothermal hot-compression tests at 650 °C and 5 s^−1^ with the 0.55 strain are carried out, and the comparison between the measured flow curves and predicted flow stresses at this deformation conditions is displayed in Figure 5. The predicted flow stresses based on the modified Arrhenius-type model by the proposed approach have a better accordance with measured flow curves compared with the other three developed constitutive models. The abovementioned analyses indicate that the modified Arrhenius-type model by the proposed approach exhibits a more accurate prediction of flow behavior than the other three developed constitutive models for highly purified Fe-Cr alloy in this study.

In addition, during the prediction of flow stress, the numbers of parameters that need to be determined besides strain, strain rate and temperature are 0, 3, 5 and 3 for the modified JC model by conventional approach and by considering strain dependency of the parameters, modified Arrhenius-type model by conventional approach and by the proposed approach, respectively. This indicates that the required calculations for describing of the flow behavior according to the modified Arrhenius-type model developed by the proposed approach are relatively simple compared to those based on modified Arrhenius-type model developed by conventional approach. Moreover, the modified Arrhenius-type model by the proposed approach possesses better performance than that for metals having only ferrite by conventional approach [16,17,33,47,48] (Figure 6). Therefore, the modified Arrhenius-type model by the proposed approach is suggested to perform constitutive description for metals having only ferrite during deformation.

### 3.3. Dynamic Microstructure Analysis

The microstructure analyses are the powerful tool to further elucidate the dynamic softening mechanism, understand the microstructure evolution during deformation and hence optimize rolling process and improve formability and surface quality of Fe-Cr alloys. In order to elucidate the microstructure evolution and dynamic softening mechanism in-depth, the dynamic microstructures at different deformation conditions were systematically analyzed by using EBSD or/and TEM. The activation energy *Q* acquired in Section 3.2.4. was used to calculate *Z* (presented in the following figures) for establishing the relationship between microstructure evolution and deformation conditions. Figure 7 shows EBSD analyses of the dynamic microstructures at different deformation conditions. These dynamic microstructures consist of many elongated grains and some fine equiaxed grains formed at the grain interiors and along the original high-angle grain boundaries (HAGBs) (marked by the white arrows in the Figure 7c–e), which are fully developed during deformation. This indicates that DRX occurs besides extensive DRV.

The SFE of steels were usually determined by the following methods [49,50,51,52]: (1) the calculations by thermodynamic modeling or empirical formula; and (2) the measurements by analyzing the structure of partial dislocations and their distance through TEM or using the XRD line broadening and diffraction line displacement analysis through X-ray diffraction. These routine methods are mainly based on the fcc metals, which cannot be applicable for calculating or measuring the experimental steel with bcc structure. Although the SFE of experimental steel is not obtained in this study, it is well-known that, as with Al alloys (~166 mJ·m^−^^2^) and Nickel (128 mJ·m^−2^), Fe-Cr alloy or ferritic stainless steel also possesses high SFE [51,52]. In this case, dislocation cross-slip can easily occur due to high SFE [13], which is beneficial to reducing the dislocation density and subsequently hindering the occurrence of full DRX. Therefore, fairly efficient DRV and weak DRX occurs during hot deformation and the microstructures after deformation consist of many elongated grains and some new grains.

For the formation of DRX grains, there are some significant differences at different deformation conditions. At lower *Z*-deformation conditions, all the DRX grains can be randomly formed along the original HAGBs and in the grain interiors. The number of new grains formed along the original HAGBs increases and that nucleated in the grain interiors decreases, and the DRX grain size increases, as the temperature rises (Figure 7c,d). At higher *Z*-deformation conditions, almost all the newly formed grains can be observed along the in-grain shear bands (Figure 7e). The number of DRX grains increases in the deformed grains, when the number of shear bands increases.

#### 3.3.1. Nucleation Mechanism of DRX Randomly Occurred along The Original Hagbs

In order to elucidate the random formation of DRX grains along the original HAGBs during deformation with lower *Z*, the microstructure evolution along the grain boundary was analyzed, as shown in Figure 7 and Figure 8. At the initial stage of deformation, the serrated grain boundary is formed (Figure 7b). Misorientation analysis was implemented near the serrated grain boundary along the arrows A1 and A2 in Figure 7b for analyzing the change of orientation gradients and stored energy (Figure 8). The local (point-to-point) misorientations are less than 2°, but the cumulative (point-to-origin) misorientations gradually increase near the serrated grain boundary. Moreover, the cumulative misorientation formed along the arrows A1 shows a larger orientation gradient than that along the arrows A2, indicating the stored energy difference across the serrated grain boundary. With the further deformation, grain boundary bulging is developed, and subgrain boundaries are regularly observed before some bulged regions (marked by the black long arrows in Figure 7c). The analysis of misorientation was also implemented behind bulged regions along the arrows B1 and B2 in Figure 7c (Figure 8). Behind bulged regions of the original HAGBs, a large orientation gradient can be formed. This large orientation gradient behind bulged regions is a beneficial condition for the formation of subgrain boundary [53,54]. Subgrain boundaries are also formed behind some bulged regions and part of subgrain boundaries transform into HAGBs (marked by the black short arrow in Figure 7c), and the DRX nuclei completely composed of HAGBs are randomly formed along the original HAGBs (marked by the white arrow in Figure 7c).

For the DRX grains randomly nucleated along the original HAGBs during deformation with lower *Z*, the mechanism is clarified in the schematic representation in Figure 9 according to the abovementioned microstructure evolution. During deformation, the serrated grain boundary is formed due to the incompatibilities between grains, which can hinder the grain boundary sliding and promote the dislocation accumulation, eventually resulting in the formation of local high dislocation density gradients along the original HAGBs (Figure 9a). This is observed in Figure 7b and Figure 8a,b. Due to the grain boundary migration driven by the stored energy difference, some part of the original HAGBs migrates to the region with high dislocation density, leading to the grain boundary bulging; and meanwhile, at the region with high dislocation density gradients near the original HAGBs, subgrain boundaries are formed, owing to the annihilation and rearrangement of dislocations (Figure 8b), which can be confirmed by the observation marked by the black long arrows in Figure 7c. As the deformation continues, grain boundary sliding results in the local concentration of strains and high dislocation density behind the bulged regions for accommodating the plastic strain, in agreement with the misorientation analysis in Figure 8c,d and subsequently subgrain boundaries are formed behind the bulged regions through the dislocation migration (Figure 9c), proved by the observation in Figure 7c. With the further deformation, dislocations accumulate in the subgrain boundaries behind the bulged regions, leading to the increase of their misorientations. When these misorientations become enough large, the DRX nuclei developed from the grain boundary bulging are separated from the original HAGBs grains and the DRX grains are formed along the original HAGBs (Figure 9d). These correspond with the observation marked by the black short arrow and white arrow in Figure 7c. These results and analyses indicate that, during deformation with lower *Z*, the DRX grains along the original HAGBs can be nucleated by the grain boundary bulging mechanism; that is to say, the DDRX is responsible for the random formation of DRX grains along the original HAGBs during deformation with lower *Z*. A similar phenomenon has been found during the hot/warm deformation of other ferritic steel [10] and austenitic steel having flow curves which are similar to that of the experimental steel [34].

According to the abovementioned nucleation mechanism of DRX, it can be concluded that grain boundary migration has an obvious effect on the DRX grains nucleation that randomly occurred along the original HAGBs during deformation, with lower *Z* and parameters affecting the grain boundary migration also inevitably affecting the number of DRX grains. The velocity of grain boundary migration (*V*) can be expressed as follows [55]:*V = M·P*(14)
where *P* is net pressure on the grain boundary; *M* is the boundary mobility related to temperature (*T*) by following an Arrhenius-type of relationship, *M* = *M*_0_exp(−*Q*_a_/*RT*); *Q*_a_ is the apparent activation energy. Increasing temperature can accelerate grain boundary migration and then the DRX grain formation. Therefore, the number of DRX grains randomly formed at the original HAGBs increases as the temperature rise (Figure 7c,d); that is, the effect of DDRX characterized by grain boundary bulging became stronger.

#### 3.3.2. Nucleation Mechanism of DRX Randomly Occurred in The Original Grain Interiors

In order to elucidate the random formation of DRX grains in the original grain interiors during deformation with lower *Z*, the microstructure evolution in the grain interiors was analyzed, as shown in Figure 10 and Figure 11. At the initial stage of deformation, a large number of dislocations are generated (Figure 10c). Some subgrain boundaries are formed or are being developed due to the dislocation mobility (marked by white and red arrows in Figure 10c, respectively). The original grain is subdivided by these subgrain boundaries, which is due to the changes of local orientations in original grain produced by the strain incompatibilities between neighboring grains [53,56]. With the further deformation, the subgrain boundaries become more distinct, and their number increases (Figure 10d). Moreover, some subgrain boundaries will grow into the regions with high dislocation density (marked by red arrow in Figure 10d) and absorb the dislocations in these regions (as an insert in Figure 10d and marked by white arrow). Finally, the DRX nuclei completely composed of HAGBs are formed in the original grain interiors (marked by the white arrow in Figure 7d). Furthermore, misorientation distributions of grain boundaries in the interiors of the original deformed grains at different strains (no including original HAGBs) were analyzed during deformation with lower *Z*, according to the Figure 7a–c (Figure 11). The proportion of HAGBs and average misorientation increase in the original grain interiors with increasing strain. In other word, the misorientation of the grain boundaries in the grain interiors increases with the continuous occurrence of deformation, which corresponds to the results of TEM observation.

For the DRX grains randomly nucleated in the original grain interiors during deformation with lower *Z*, the mechanism is clarified in the schematic representation in Figure 12 according to the abovementioned microstructure evolution. During deformation, a large number of dislocations are produced on slip planes (Figure 12a), which subsequently transform into subgrain boundaries due to the dislocation cross-slip and climb (Figure 12b). This can be demonstrated by the observation marked by white and red arrows in Figure 10c. As the deformation continues, the subgrain boundaries become more distinct and their number increases through the annihilation and rearrangement of dislocations (Figure 12c), in agreement with the TEM observation in Figure 10d. Moreover, subgrain boundaries will begin to migrate into the regions with high dislocation density, driven by the stored energy difference (marked by the red arrow in Figure 12c). This is confirmed by the observation marked by red arrow in Figure 10d. During the migration of subgrain boundaries, dislocation is progressively trapped into these subgrain boundaries, leading to the uniform increase in their misorientation and their transformation into HAGBs, and finally, the DRX grains are formed in the original grain interiors (Figure 12d). These are in line with the observation in an insert in Figure 10d, misorientation distribution in Figure 11 and the formation of new grains marked by the white arrow in Figure 7d. These results and analyses indicate that during deformation with lower *Z*, the DRX grains in the original grain interiors can be nucleated by the uniform increase in misorientation between subgrains; that is to say, the CDRX is responsible for the random formation of DRX grains in the original grain interiors during deformation with lower *Z*.

According to the abovementioned nucleation mechanism of DRX, it can be concluded that subgrain boundaries can provide sites for the nucleation of DRX grains randomly formed in the original grain interiors during deformation with lower *Z*, and parameters affecting the formation and migration of subgrain boundaries will also inevitably affect the number of DRX grains. According to Section 3.1, increasing deformation temperature can facilitate the activation of new slip systems, grain boundary sliding and migration and dislocation mobility. This can suppress the formation of subgrain boundary by decreasing the dislocation density, impede the migration of subgrain boundary and the increase its misorientation by hindering the generation of regions with high dislocation density in the grain interiors, and eventually inhibit the formation of DRX grain. Therefore, the number of new grains randomly formed in the original grain interiors decreases as the temperature increases (Figure 7c,d); that is, the effect of CDRX induced by increasing the misorientation of subgrain boundaries became weaker. A similar sensitivity of the effect of CDRX to deformation temperature has been observed by other researchers [21].

During deformation with lower *Z*, as the temperature rises, the DRX grain size increases (Figure 7c,d), indicating that DRX becomes more active. The size of DRX grain is closely related to the DRX nuclei size during its nucleation and grain boundary migration during its growth. For the nucleation of DRX grain by the grain boundary migration and bulging, Mandal et al. [55] suggested that the critical radius of DRX nucleus *r*_c_ is related to the stored energy in a grain *U* and can be expressed as follows:*r*_c_ = 2*γ*_b_·*U*^−1^(15)
where *γ*_b_ is boundary energy. As the stored energy is lower, the critical radius of DRX nuclei is larger. Generally, the stored energy is low during deformation at higher temperature. Hence, the size of DRX nuclei formed by the grain boundary migration and bulging is larger with the increase of deformation temperature. On the other hand, for the formation of DRX nuclei by uniform increase in misorientation of subgrain boundaries, the transformation of subgrain into DRX nucleus needs that a certain number of dislocations are trapped into the subgrain boundaries. According to Section 3.1, however, the dislocation density is lower after deformation at a higher temperature. Thus, during deformation at a higher temperature, the subgrain boundary needs to migrate into the larger regions, and then the dislocations trapped into the subgrain boundaries can reach a critical value, resulting in the transformation of subgrain into DRX nucleus; in other words, the size of DRX nuclei formed by increasing the misorientation of subgrain boundaries increases with the deformation temperature. During the growth of DRX grain, the DRX grain size is closely related to the boundary migration. According to Equation (14), increasing the deformation temperature can accelerate the grain boundary migration, promoting the growth of DRX grain. Therefore, the grain size increases with the deformation temperature for the DRX grains randomly formed along the original HAGBs and in the grain interiors during deformation at lower *Z*.

#### 3.3.3. Nucleation Mechanism of DRX Occurred along The In-Grain Shear Bands

At higher *Z*-deformation conditions, a large number of shear bands are also formed along with the formation of many new grains (Figure 7e). As stated in Section 3.1, decreasing the deformation temperature can decrease the strain rate sensitivity. Due to the different strain partitioning during deformation, the deformation of some local areas may be severe compared with the deformation of whole sample. In this case, these local areas possess high strain rate compared with the whole sample. However, in these local areas, the stress rise is very small or even reduced due to lower strain rate sensitivity, especially when the experimental steel is deformed at a relatively low temperature, which can promote flow localization. This indicates that relatively low strain rate sensitivity could increase the probability of flow localization. Furthermore, it was reported that the factors that increase the probability of flow localization could promote the formation of in-grain shear band [33]. Therefore, the lower deformation temperature can enhance the tendency for the occurrence of in-grain shear band; that is to say, a large number of shear bands can be easily formed after deformation at relatively high *Z*.

Moreover, the number of shear band families developed along various directions, and the density of the shear bands are apparently different for different grains. In some grains, two sets of shear bands intersecting at about 50° are clearly observed, only one family of shear bands can be formed in some grains and there are no shear bands in some other grains. This inhomogeneous distribution of shear bands can probably play a significant role in the number of DRX grains in different original deformed grains (the number of DRX grains increases with the number of shear bands). During deformation, the appearance of more family of shear bands and higher density of shear bands indicates the activation of multiple shearing. An operation of multiple shearing is a beneficial condition for generating strain-induced (sub)grain boundaries and dynamic recrystallization during deformation [56].

In order to explore the nucleation of new fine grain during deformation with higher *Z*, the dislocation, in-grain substructure and misorientation distribution of grain boundaries in the deformed grain interiors were analyzed, as shown in Figure 13 and Figure 14. Note that the misorientations of some strain-induced (sub)grain boundaries in Figure 13 were analyzed by the PED and indicated in the corresponding (sub)grain boundaries. During deformation, the microbands, which are elongated cell structures containing some dislocation tangles, are formed. These microbands are long, thin plate-like features formed on slip planes within the original deformed grains, and their boundaries were characterized by one set of dense dislocation walls (Figure 13a,b), which is similar in appearance to the microbands that are observed in other metals during deformation [3,33,57]. In localized regions, the larger strain can lead to the tangles and dislocation networks along different direction from microbands (as shown in region A in Figure 13b), and, subsequently, one set of shear bands is observed (Figure 13c), which is in agreement with the shear band that formed in the warm-deformed microstructures of other metals [3,33,56]. These shear bands are divided into cell blocks by the subgrain boundaries. At the shear band, some subgrains have sharp subgrain boundaries and many of subgrain boundaries have high misorientation, whereas, in other regions, the strain-induced subgrain boundaries are scattered and possess relatively low misorientation (Figure 13c).

With the further increase of strain in some localized regions, the misorientations of subgrain boundaries rapidly increase, leading to the formation of some strain-induced HAGBs (Figure 13d). Meanwhile, the shear bands along another new direction are formed, and then two sets of mutually intersected shear bands are observed (Figure 7e). Based on the misorientation analysis in Figure 7e and Figure 13, it can be seen that, in the shear bands, the strain-induced subgrain boundaries have high misorientation, and some HAGBs are formed, especially at their intersections and boundaries, while the strain-induced subgrain boundaries in the regions located far from the shear band usually possess low misorientation. This indicates that the increase of misorientation between subgrains cannot occur homogeneously in the grain interiors, and it mainly takes place within the shear bands, especially at their intersections and boundaries; that is to say, the shear bands are the underlying and beneficial sites for the formation of HAGBs and new grain.

The misorientation distributions of grain boundaries in the interiors of original deformed grains having different shear hand families and density in Figure 7e were analyzed, as shown in Figure 14. It can be seen that, for deformed grains without shear bands, the misorientation distribution is dominated by the high fraction of LAGBs which is related to a large number of dislocation cells (Figure 13a). After the formation of shear bands, the fraction of HGABs and the average misorientation increase with the number of shear band families and density of shear bands. This continuous increase in misorientation between subgrains can be duo to the formation of new fine grains along the shear bands, and it is an essential feature of CDRX. In fact, this continuous increase in misorientation mainly takes place at the shear band, namely in some local regions of microstructure. It is different from the progressive increase in misorientation occurring during the nucleation of DRX grains randomly formed in the original grain interiors. The latter can take place in any region, thereby resulting in the random formation of DRX grains in the original grain interiors, and belongs to the homogeneous and extensive increase.

In addition, Sakai et al. [56] suggested that the strain-induced HAGBs are scarcely developed at the strain range of 0 to 1.5 during warm deformation. In this study, however, after deformation at 700 °C to 0.55 strain, many HAGBs were formed in some of the deformed grains. This is due to the strengthening of flow localization and the obvious increase of strain in the localized regions during deformation at relatively high *Z*. It is very difficult to measure the strain of each grain, but it can be qualitatively evaluated through Kernel average misorientation (KAM) according to the EBSD measurement. KAM can give evidence for the stored energy in the deformed microstructure, and high KAM values indicate the high stored energy and the large strain. Through EBSD analysis (Figure 7e), the KAM in some grains is significantly higher than other grains, as shown in Figure 15. This indicates that the strain of these grains is much larger than the strain of matrix (0.55), leading to the formation of strain-induced HAGBs and new fine grains. This further confirms that the gradual increase in misorientation between subgrains with the number of shear bands mainly occurs in a localized region, rather than any region.

In this section, the fraction of new grains and misorientations of strain-induced subgrain boundaries continuously increase with the number of shear band families and density of shear bands during deformation. This is the feature of microstructure evolution during CDRX. For these new grains formed along the in-grain shear bands during deformation with higher *Z*, the nucleation mechanism is clarified in the schematic representation in Figure 16, according to the abovementioned microstructure observation. During deformation at relatively high *Z*, high-density dislocations are generated, and then the original grains are homogeneously divided into long, thin microbands containing dislocation tangles by the strain-induced subgrain boundaries. This is observed in Figure 13a,b. Owing to the enhancement of flow localization, the larger strain occurs in some localized regions. It leads to the formation of one set of shear bands by activating the new slip systems and eventually increase the misorientation of strain-induced subgrain boundaries due to promoting the crystal rotation of localized regions [5] (Figure 16a), in agreement with the TEM observation in Figure 13c.

With the further deformation, on one hand, some subgrain boundaries at the shear bands, especially at the boundaries of shear bands can rapidly increase their misorientations at relatively small strains [11], leading to the appearance of many strain-induced HAGBs (Figure 16b). On the other hand, the re-occurrence of localized larger strain accelerates the activation of other new slip systems and then develops another shear band family along another direction, leading to the intersection of shear band (Figure 16b). At the intersection of shear band, the activation of multiple slip systems occurs. Dorner et al. [58] suggested that the local change of active slip systems in the shear bands was a beneficial condition for crystal rotations that increases the misorientation of strain-induced subgrain boundaries and promotes their transformation into HAGBs. Thus, the HAGBs in the grain interiors are formed preferentially at the shear bands, especially at their intersections and boundaries (Figure 16b), promoting the nucleation of new grains. This is confirmed by the TEM observation in Figure 13d and EBSD analysis in Figure 7e.

With the continuous deformation, especially in localized regions, the abovementioned process is repeated, and the density of shear bands and the number of intersection of shear bands increase, promoting the increase in the fraction of HGABs. In addition, dynamic recovery during deformation would be another important condition for the formation of new grains by accelerating the motion of dislocation in the shear bands and increasing the misorientation of strain-induced subgrain boundaries. Ultimately, the new grains are formed along the shear band (Figure 16b), and its fraction increases with the number of shear band families and density of shear bands. These are demonstrated by the misorientation distribution in Figure 14 and EBSD analysis in Figure 7e. The abovementioned results and analyses indicate that, during deformation with higher *Z*, the larger plastic strain in localized regions due to the enhancement of flow localization results in the formation and intersection of shear bands, and the new grains are formed by the shear band-induced mechanism and the microstructure evolution and DRX grains formation possesses the feature of CDRX.

Through the observation of microstructures above, the different DRX nucleation mechanisms activated during deformation are analyzed, as shown in Figure 17. It can be seen that the nucleation mechanisms of new grains can be considered as a function of deformation conditions. At higher *Z*-deformation conditions (*Z* ≥ 5.93 ± 2.96 × 10^27^ s^−1^) CDRX induced by shear band occurs for the formation of new grains. At lower *Z*-deformation conditions (*Z* ≤ 3.10 ± 0.72 × 10^25^ s^−1^), the nucleation of new grain can be attributed to the combination of DDRX induced by grain boundary bulging and CDRX induced by a uniform increase in misorientation between subgrains. Moreover, according to the discussion in Section 3.3.1 and Section 3.3.2, there is a competition between CDRX and DDRX mechanisms depending on deformation temperature. With the increase of deformation temperature, the effect of DDRX induced by grain boundary bulging became stronger, whereas the effect of CDRX induced by uniform increase in misorientation became weaker. Note that the relative error of the critical *Z* parameters for the occurrence of DRX is determined by the minimum deviations between the *Z* values of the activation of DRX and no DRX, based on the limited deformation conditions. In addition, the microstructure evolution may be applicable to other Fe-Cr alloys and ferritic stainless steels. The critical deformation conditions for the operation of DRX mechanism and the activation energy of hot deformation may be different due to the different chemical compositions between experimental steel and other materials, leading to the different critical Zener-Hollomon parameters for the occurrence of DRX mechanism. However, for other Fe-Cr alloys and ferritic stainless steels, the relationship between DRX mechanism and deformation condition is similar to that of experimental steel in this study. That is to say, the CDRX induced by shear band will be prone to occur at higher *Z* d-formation conditions, while at lower *Z*-deformation conditions, DDRX induced by grain boundary bulging and CDRX induced by uniform increase in misorientation will easily take place and with the increase of temperature, the effect of DDRX became stronger, whereas the effect of CDRX became weaker.

## 4. Conclusions

The flow behavior and restoration mechanisms resulting in the dynamic microstructure evolution of highly purified Fe-17Cr alloy were investigated by using compression tests. The following conclusions can be drawn:

(1) Deformation temperature and strain rate play a key role in flow stress, and increasing temperature and decreasing strain rate lead to decreasing flow stress. Deformation temperature has a significant effect on strain-rate sensitivity (stress-increasing tendency with the increase of strain rate), and as temperature lowers, strain-rate sensitivity reduces, which can be explained in terms of the stacking fault energy.

(2) During deformation, dynamic recrystallization occurs alongside extensive dynamic recovery, and the nucleation mechanisms of new grains can be considered as a function of deformation conditions. At higher *Z*-deformation conditions (*Z* ≥ 5.93 × 10^27^ s^−1^), lower strain-rate sensitivity could increase the probability of flow localization, thereby promoting the formation of in-grain shear band, and continuous dynamic recrystallization induced by shear band can be responsible for the formation of new grains. At lower *Z*-deformation conditions (*Z* ≤ 3.10 × 10^25^ s^−1^), the nucleation of new grain can be attributed to the combination of continuous dynamic recrystallization induced through a uniform increase in misorientation between subgrains and discontinuous dynamic recrystallization induced through grain boundary bulging.

(3) At lower *Z*-deformation conditions (*Z* ≤ 3.10 × 10^25^ s^−1^), there is a competition between continuous and discontinuous dynamic recrystallization mechanisms as a function of deformation temperature, and with the increase of temperature, the effect of discontinuous dynamic recrystallization induced by grain boundary bulging became stronger, whereas the effect of continuous dynamic recrystallization induced by the uniform increase in misorientation became weaker. Moreover, the DRX grain size increases with the deformation temperature.

(4) The modified Arrhenius-type model was constructed by using the proposed approach through distinguishing stress levels for acquiring partial parameter, employing peak stress to determine the activation energy and considering strain dependency of only other parameters for compensating strain. The modified Johnson-Cook model, by considering strain dependency of the parameters, and the modified Johnson-Cook and Arrhenius-type models, by conventional approach, were also developed.

(5) The modified Arrhenius-type models by the proposed approach can all offer a more accurate prediction of flow behavior under investigated strain rate–temperature domain as compared to the other three developed constitutive models in this study. The modified Arrhenius-type model by the proposed approach is recommended to describe the flow behavior for alloys and steels having only ferrite during deformation, especially at lower temperatures. In addition, for hindering the surface ridging of Fe-Cr alloys, it seems advantageous to activate the CDRX induced by shear band and form a considerable number of new fine grain nuclei through hot deformation with higher Z, because this is beneficial to enhancing grain fragmentation and grain interior nucleation during annealing, thereby weakening the inhomogeneous orientation distribution.

## Figures and Tables

**Figure 1 materials-14-05022-f001:**
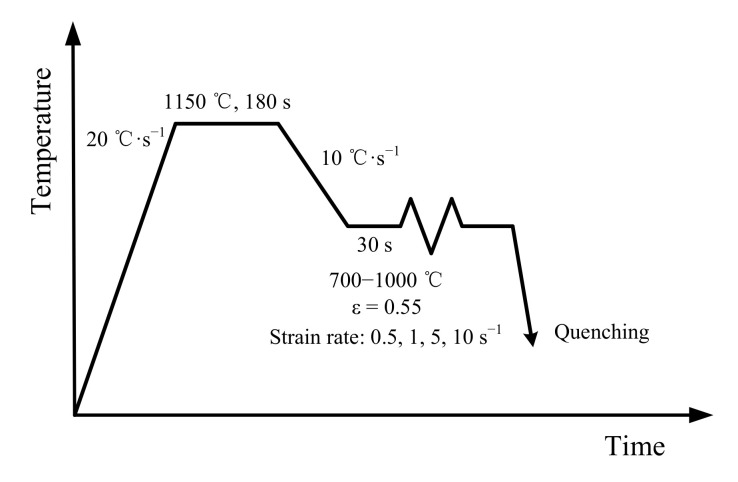
Hot-deformation processes.

**Figure 2 materials-14-05022-f002:**
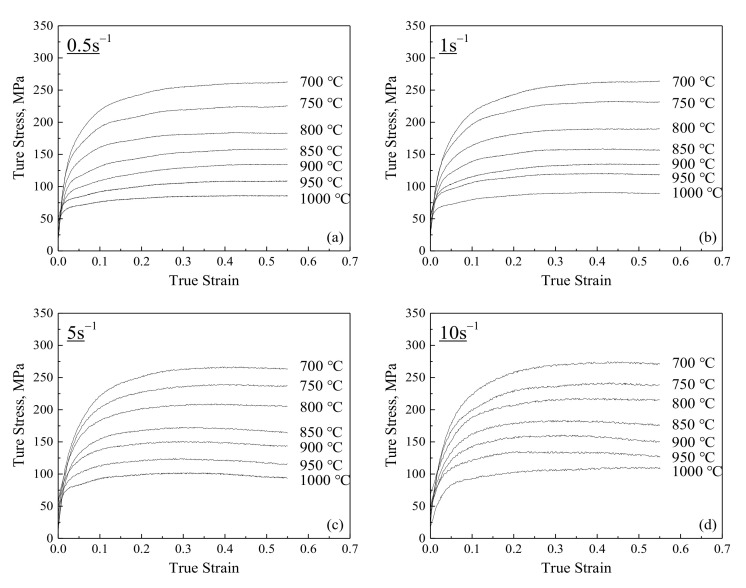
Flow-stress curves under different processes: (**a**) 0.5 s^−1^, (**b**) 1 s^−1^, (**c**) 5 s^−1^ and (**d**) 10 s^−1^.

**Figure 3 materials-14-05022-f003:**
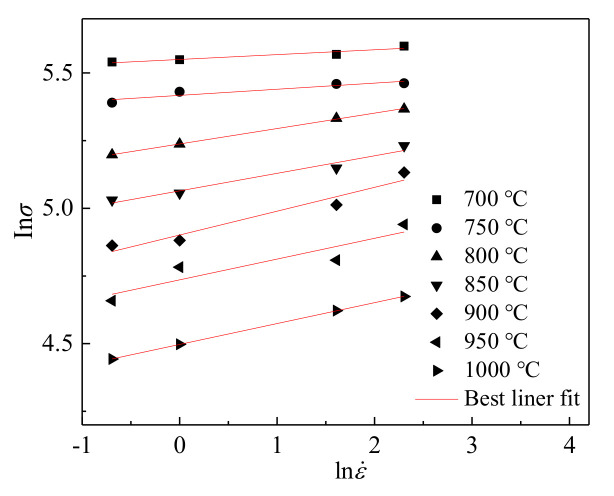
Relationship between flow stress and strain rate for alloy deformed at 900 °C to 0.3.

**Figure 4 materials-14-05022-f004:**
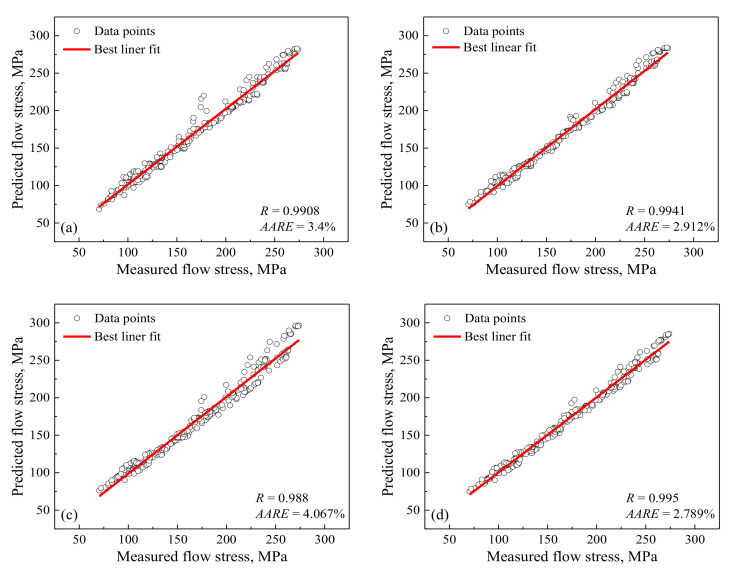
The *R* and *AARE* between the measured and calculated values based on different constitutive models: (**a**) the modified JC by conventional approach, (**b**) modified JC by considering strain dependency of the parameters, (**c**) modified Arrhenius-type model by conventional approach and (**d**) modified Arrhenius-type model by the proposed approach.

**Figure 5 materials-14-05022-f005:**
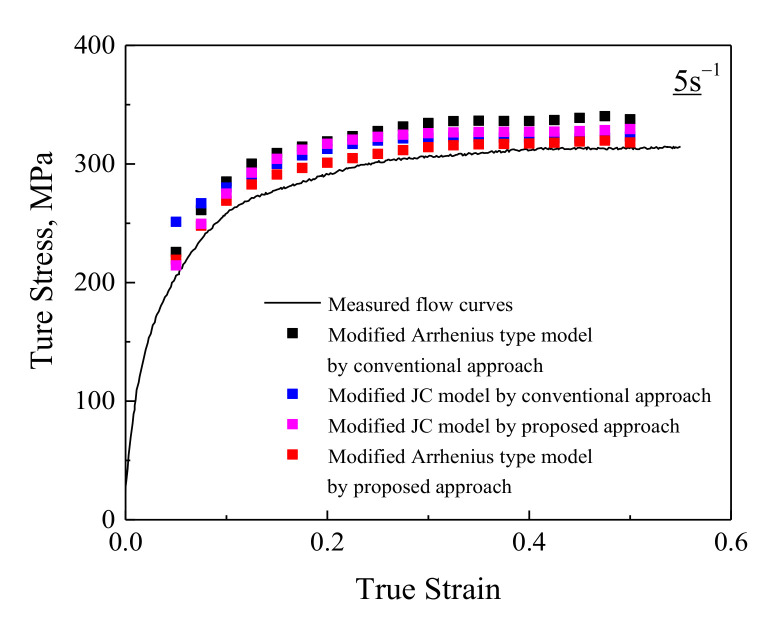
Evaluation of the developed constitutive models through the measured data after deformation at the 650 °C and 5 s^−1^.

**Figure 6 materials-14-05022-f006:**
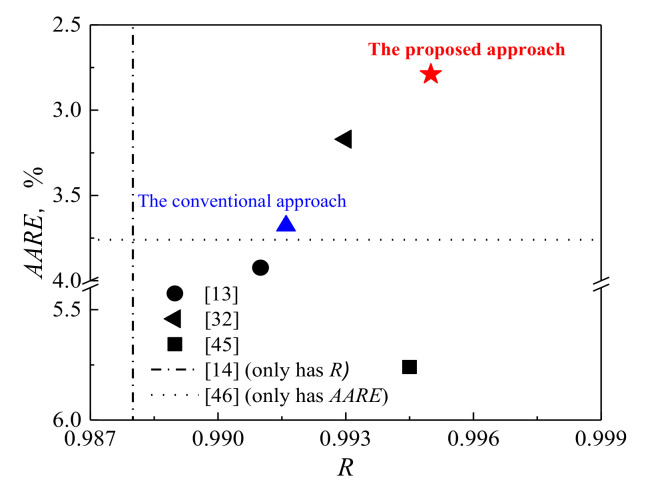
The *R* and *AARE* for modified Arrhenius-type models constructed by the proposed approach in present work and by other studies.

**Figure 7 materials-14-05022-f007:**
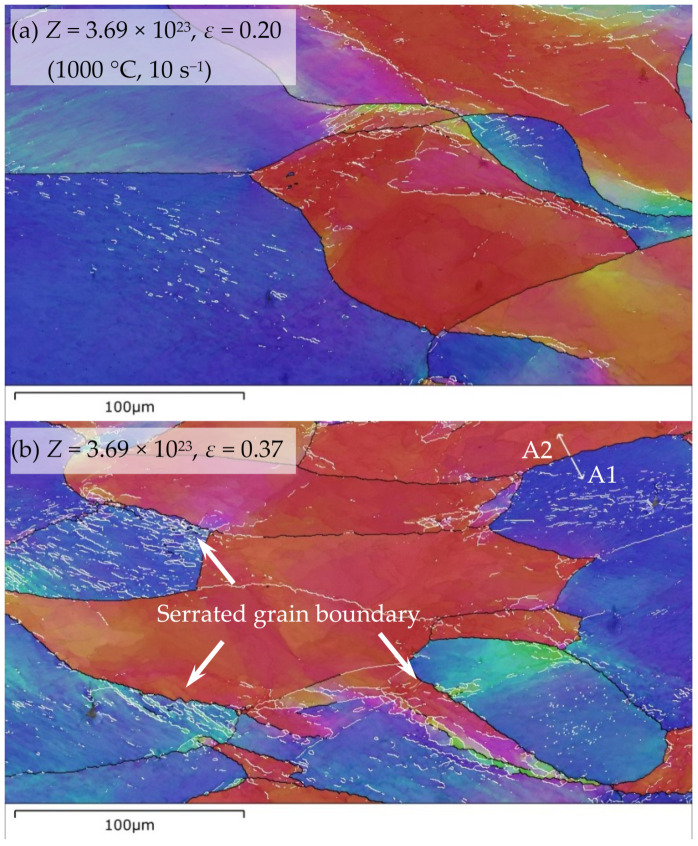
EBSD analyses of the dynamic microstructures (inverse pole figure maps for the CA (compression axis) direction) after deformation at different deformation conditions. (White and black lines denote low-angle grain boundary (LAGBs) (2–15°) and high-angle grain boundary (HAGBs) (greater than 15°), respectively).

**Figure 8 materials-14-05022-f008:**
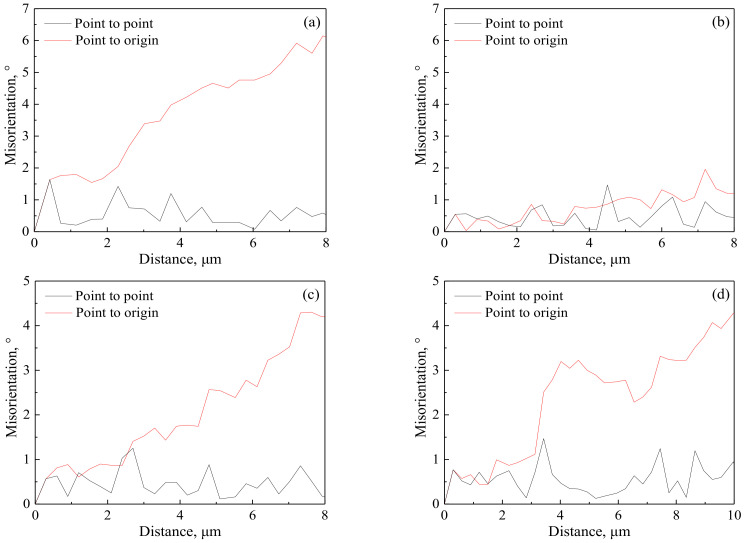
Misorientation angle changes near the original high-angle grain boundaries along the arrows in Figure 7b,c: (**a**) A1, (**b**) A2, (**c**) B1 and (**d**) B2.

**Figure 9 materials-14-05022-f009:**
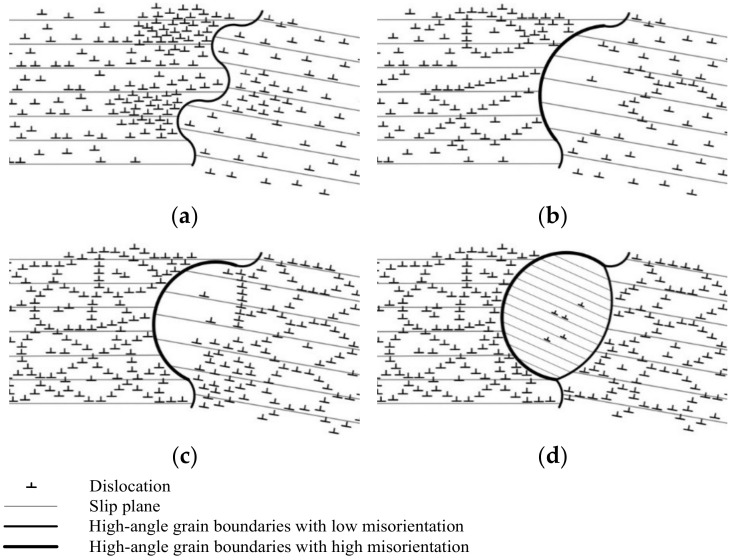
Schematic diagram showing the nucleation model of DRX randomly occurred along the original high-angle grain boundaries by grain boundary bulging. (Note that only dislocations of one sign are shown in the schematic diagram for simplifying model.) (**a**) The formation of serrated grain boundary. (**b**) Grain boundary bulging. (**c**) The formation of subgrain boundary behind the bulged regions. (**d**) DRX grain nucleation.

**Figure 10 materials-14-05022-f010:**
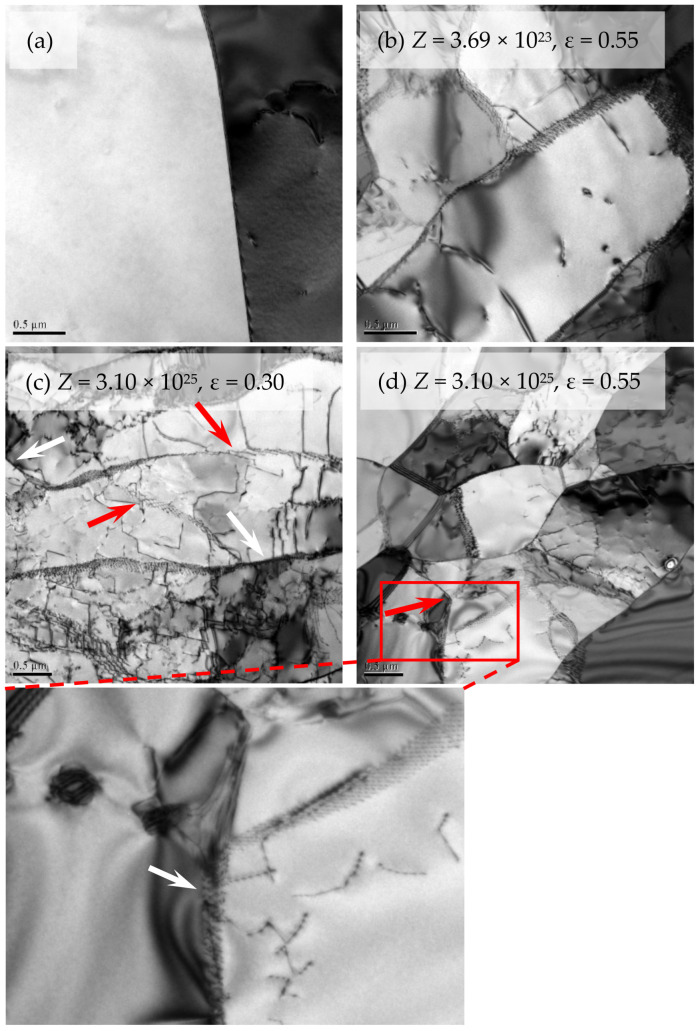
TEM analyses of the initial microstructure (**a**) and dynamic microstructures (**c**–**d**) after deformation at different deformation conditions.

**Figure 11 materials-14-05022-f011:**
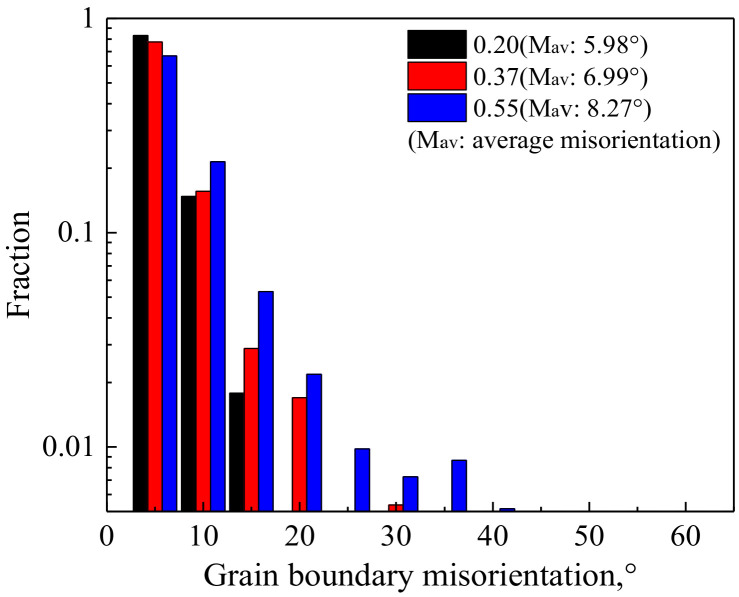
Misorientation distribution of grain boundaries in the interiors of all the original deformed grains (not including original high-angle grain boundaries) after deformation at *Z* = 3.69 × 10^23^ to different strains.

**Figure 12 materials-14-05022-f012:**
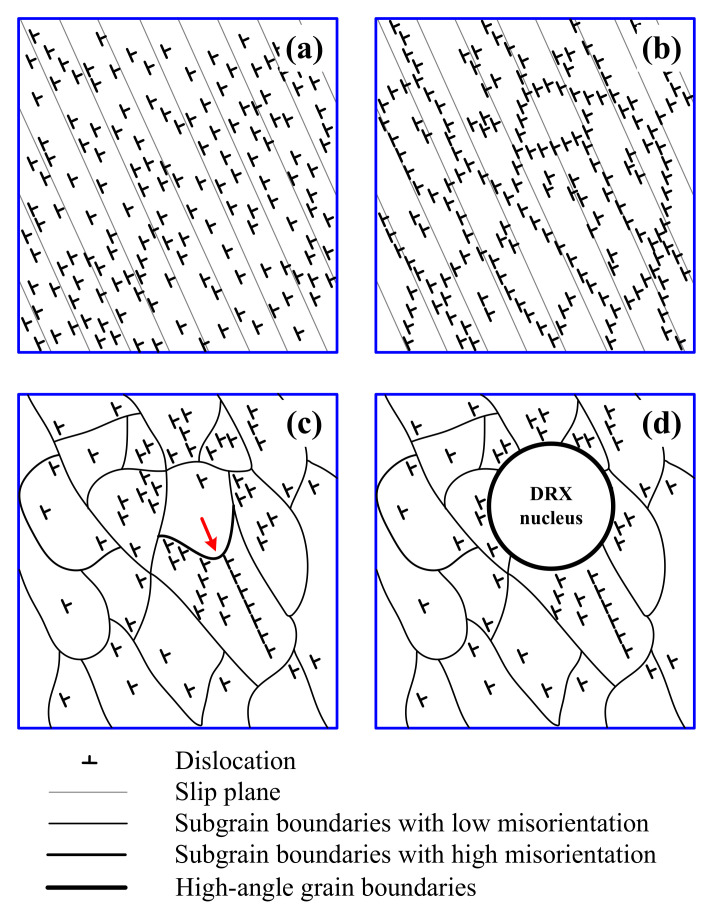
Schematic diagram showing the nucleation model of DRX randomly occurred in the original grain interiors by uniform increase in misorientation between subgrains. (Note that only dislocations of one sign are shown in the schematic diagram for simplifying model.) (**a**) Dislocation generation, (**b**) subgrain formation, (**c**) subgrain boundary migration and (**d**) DRX grain nucleation.

**Figure 13 materials-14-05022-f013:**
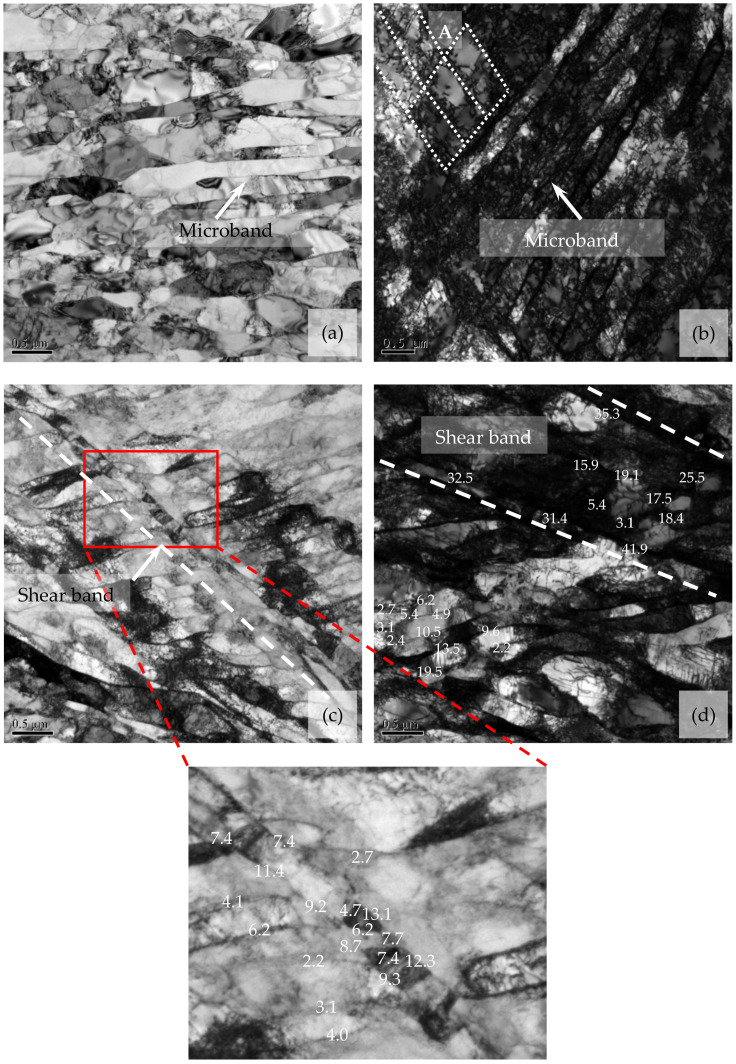
TEM analyses of the dynamic microstructures after deformation at *Z* = 3.34 × 10^30^ and strain of 0.55 (The insert in Figure 13c shows the misorientation analyses of some strain-induced subgrain boundaries at and near the shear band. The numbers indicate the misorientation angles between subgrains obtained by using precession electron diffraction (°)).

**Figure 14 materials-14-05022-f014:**
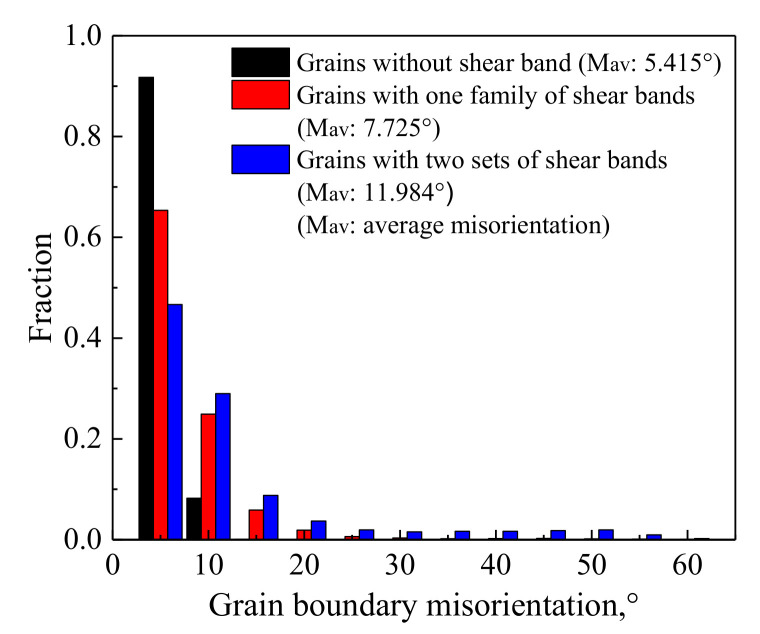
Misorientation distribution of grain boundaries in the interiors of the original deformed grains having different shear hand families and density (not including original high-angle grain boundaries) after deformation at *Z* = 3.34 × 10^30^ and strain of 0.55.

**Figure 15 materials-14-05022-f015:**
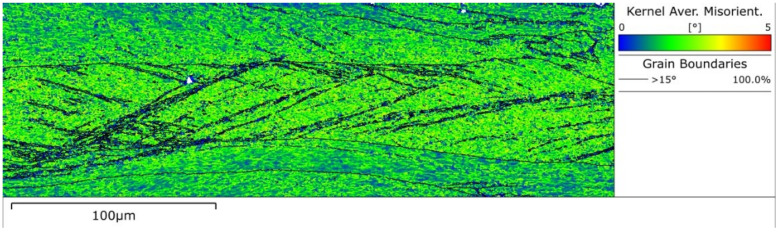
Kernel average misorientation map of the dynamic microstructure after deformation at *Z* = 3.34 × 10^30^ (700 °C, 10 s^−1^) and strain of 0.55. (Black lines denote high-angle grain boundary having the misorientation greater than 15°.)

**Figure 16 materials-14-05022-f016:**
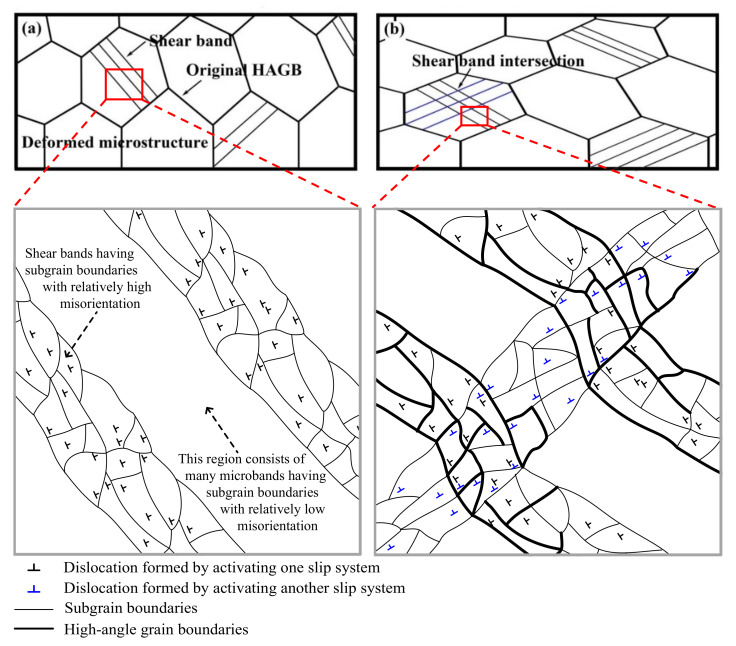
Schematic diagram showing the nucleation model of DRX formed along the in-grain shear band. (**a**) The formation of one set of shear bands. (**b**) The intersection of shear band and formation of new grains along the shear band. (Note that only dislocations of one sign are shown for one activated slip system in the schematic diagram for simplifying model.).

**Figure 17 materials-14-05022-f017:**
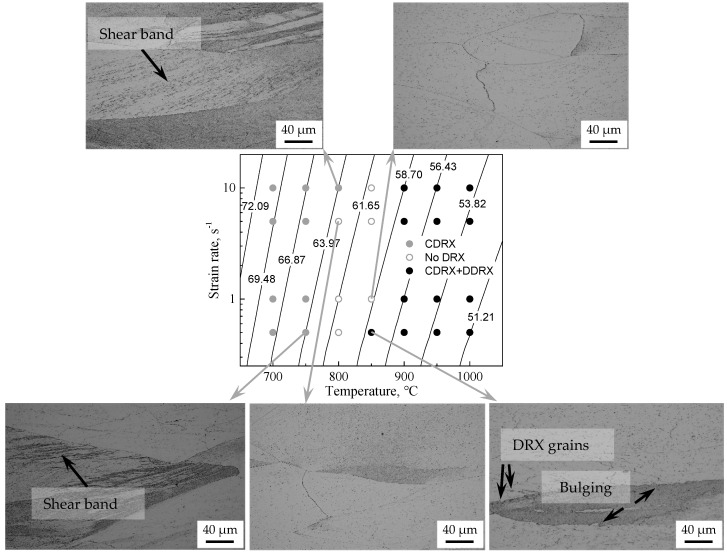
Contour map of Zener-Hollomon parameters and activation domain of the different DRX nucleation mechanisms over the entire strain rate and temperature range investigated. (The dynamic microstructures are also shown for several typical deformation conditions. The numbers indicate the value of ln*Z.*).

**Table 1 materials-14-05022-t001:** Parameters of the modified JC model at reference condition of 850 °C and 1 s^−1^.

*B* _1_	*B* _2_	*B* _3_	*B* _4_	*B* _5_	*λ* _1_	*λ* _2_
103	439	−1171	1019	0.071	−0.00342	2.3 × 10^−4^

## Data Availability

Data sharing is not applicable to this article.

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
