# Peer review of "Constitutive Descriptions and Restoration Mechanisms of a Fe-17Cr Alloy during Deformation at Temperatures of 700–1000 °C"

_materials, 2021, doi:10.3390/ma14175022_

Round 1

Reviewer 1 Report

In this paper, the deformation behavior of a high purified Fe–17Cr alloy was investigated using isothermal compression tests under a wide range of deformation temperature, such as lower deformation temperatures which cover the practical processing conditions during the warm rolling process of Fe–Cr alloys.Besides,the modified constitutive equation was developed based on the Johnson-Cook and Arrhenius-type models.

The topic is actual. The paper is of sufficient originality and is worthy of publication even if it is mandatory to check the text in order to correct some mistakes. Attached please find the manuscript with specific comments. English Improvement needed in yellow, character change needed in red (why Italic?).

Reviewer 2 Report

The article presents an interesting modelling approach combined with experimental results under a large number of different conditions.

Although the content is interesting and the methods used are valid, the presentation is very unstructured, making it more difficult to follow. Most of the content is in a "Results and discussion" section, and no distinction is made between new results and discussion. Some of the explanations given about the deformation and nucleation process might be correct, but it is not clear how they are concluded from the obtained results. Also, some of the decisions taken with respect to the modelling look quite arbitrary, and a better explanation would be deserved.

More specific comments are given below:

  • The whole section 2 is in cursive
  • It is not specified what is the loading direction in the tests (section 2), neither how many tests are performed under each condition
  • Lines 182-198: factors 1-4 are not independent. In particular, are not 2, 3 and 4 practically a consequence of 1?
  • Lines 216-243: how does this explanation correlates with the results just presented? do the results help to prove or disprove this theory?
  • There are several occurrences of "flow  ehaviour", eg: lines 254 and 256
  • Line 257: it is claimed that the reason to modify the JC model is to include the coupled effect of strain rate and temperature, why is the equation for the flow at reference conditions (first term) changed?
  • Figure 4a: it says "pionts" instead of "points" and "3th" instead of "3rd"
  • Line 308: the term "dependent constant" is a contradiction
  • Lines 391-405: it is not obvious that the peak represent a point at which work hardening levels are the same
  • Line 482: once the model parameters are represented by 6th order polynomials, it does not really use simple mathematical expressions
  • Lines 491-502: since the experimental results used for the fitting were also affected by thermal softening, why would it consistently overestimate? shouldn't the error be evenly distributed around the experimental curve?
  • Section 3.2: due to the combination of different models and dependent parameters, it is difficult to keep track of the total number of parameters used in each model. These numbers should be given
  • Sections 3.3.1 - 3.3.3: in these 3 sections, first experimental results are presented and then a theoretical explanation is given with a diagram (which, by the way, are very clear and nicely done). It should be more clear how both correlate, with clear indications of how the experiments presented sustain the presented hypothesis or what kind of experiments could settle the issue
  • Lines 859-860: was the strain of the grains measured? what was the maximum value observed?
  • Lines 917, 919 (and conclusions): very precise limits are given for the Z parameter that determines the nucleation mechanism. How accurate are they? Shouldn't they have a relative error? Could anything be said about the possibility of applying them to other materials or how they would change?

Finally, the article needs to be carefully revised to fix grammar and spelling mistakes. Some examples from the first pages: in generally (line 106), process instead of process (for example in line 109), in the literatures (line 165), own similar shape (lines 172-173).

Reviewer 3 Report

In the paper "Constitutive Descriptions and Restoration Mechanisms of a Fe–17Cr Alloy during Deformation at Temperatures of 700-1000℃", the authors investigate the warm/hot deformation behaviour of the binary steel using a thermomechanical simulator and microstructural investigations. The authors constructed four different strain compensated models for the description of the hot deformation behaviour. The Arrhenius-type model with separation of the data by stress value shows the highest accuracy. The authors have also carried out detailed microstructural analysis for describing the microstructure evolution during the deformation. The manuscript is well written. However, it is required to be modified accordingly to the following comments:

  1. The manuscript is too large and overload by the well-known theoretical statements (e.g. lines 182-198, 216-243 and so on). The authors should carefully reread the paper and leave only the discussion about their results.
  2. A large number of the figures related to the constructed models should be moved to the Supplementary file (Figures 4-10). These figures are additional and not be primarily required for the readers.
  3. The authors wrote that the investigated binary alloy is highly purified. However, it contains 0.18 Nb and 0.14 Si. How this amount may influence the deformation behaviour?
  4. What is the SFE value of the investigated steel? And how it may influence the microstructure evolution?
  5. Why did the authors not consider the adiabatic heating during the deformation at high strain rates? This effect has a significant influence on the stress level especially at low temperatures [10.1134/S0031918X14080031, 10.1179/026708301101510843]. As result, the obtained by the authors constants values may be wrong.
  6. Minor changes:
  • The number of digits after the dot in the values of the constants should be decreased accordingly to the errors of the fitting.
  • Some typos are in Figure 15 (“piont” should be changed to “point”).

Round 2

Reviewer 2 Report

The response of the authors are satisfactory and, in my opinion, the article is ready for acceptance.

Reviewer 3 Report

The authors have improved the manuscript accordingly to previous comments. The paper may be accepted for publication.